# Ag@Au Core–Shell Porous Nanocages with Outstanding SERS Activity for Highly Sensitive SERS Immunoassay

**DOI:** 10.3390/s19071554

**Published:** 2019-03-31

**Authors:** Yaqi Huang, Dajie Lin, Mengting Li, Dewu Yin, Shun Wang, Jichang Wang

**Affiliations:** 1College of Chemistry and Materials Engineering, Wenzhou University, Wenzhou 325035, China; 16451282265@stu.wzu.edu.cn (Y.H.); 17451282201@stu.wzu.edu.cn (M.L.); dewuyin@wzu.edu.cn (D.Y.); 2Key Laboratory of Laboratory Medicine (Ministry of Education of China), Wenzhou Medical University, Wenzhou 325037, China; 3Department of Chemistry and Biochemistry, University of Windsor, Windsor, ON N9B 3P4, Canada

**Keywords:** surface-enhanced Raman spectroscopy, immunoassay, biomarkers, Ag core–Au shell porous nanocages, galvanic replacement reaction

## Abstract

A highly sensitive immunoassay of biomarkers has been achieved using 4-mercaptobenzoic acid-labeled Ag@Au core–shell porous nanocage tags and α-fetoprotein immuno-sensing chips. The Ag@Au porous nanocages were uniquely synthesized by using an Ag core as a self-sacrificial template and reducing agent, where the slow reaction process led to the formation of a porous Au layer. The size of the remaining Ag core and surface roughness of the Au shell were controlled by adjusting the chloroauric acid concentration. The porous cage exhibited excellent surface-enhanced Raman spectroscopy (SERS) activity, presumably due to a synergetic interaction between newly generated hot spots in the rough Au shell and the retained SERS activity of the Ag core. Using α-fetoprotein as a model analyte for immunoassay, the SERS signal had a wide linear range of 0.20 ng mL^−1^ to 500.0 ng mL^−1^ with a detection limit of 0.12 ng mL^−1^. Without the need of further signal amplification, the as-prepared Ag@Au bimetallic nanocages can be directly used for highly sensitive SERS assays of other biomarkers in biomedical research, diagnostics, etc.

## 1. Introduction

Clinical measurements of biomarkers can greatly help the early diagnosis, prediction, treatment monitoring and guidance of various diseases such as cancer [1]. Being able to achieve very specific and accurate detection of biomarkers is no doubt crucial in many areas of biomedical research and medical diagnostics [2]. For example, α-fetoprotein (AFP) has been developed as a biomarker for the early diagnosis of liver cancer. Generally, the AFP level exceeding the threshold of 25 ng mL^−1^ in serum is suspected to be associated with the occurrence of liver cancer or other related diseases [3]. For the above reason, in the past decade a number of novel immunosensors have been developed for cancer biomarker detection, such as enzyme-linked immunoassay [3,4], electrochemistry immunoassay [5,6,7], fluorescence immunoassay [8,9,10] and chemiluminescence immunoassay [11,12]. Although the abovementioned immunosensors have been well elaborated and some of them have even been applied in clinical trials, the development of immunoassays with better sensitivity, higher selectivity and more simplicity for rapid cancer biomarker detection in cancer patient serum is still both highly challenging and greatly needed [13].

Surface-enhanced Raman spectroscopy (SERS) is one of the modern techniques that are capable of the ultrasensitive detection and quantification of molecules on or near the surface of plasmonic nanostructures [14,15], and has wide applications in immunoassays (SERS-based immunoassays). In the process of SERS spectroscopy, the high selectivity and resolution of SERS spectra are key factors that are superior to traditional infrared and fluorescence spectroscopy. However, whether SERS-based immunoassays can be effectively employed in clinics depends on their detection sensitivity [15]. In order to achieve high sensitivity, new technical methods of fabricating the platform have been used. For example, compared with naked Au nanoparticles, enzyme-induced silver deposition could enhance the SERS signal four times [16]. Multiple SERS tags, attaching intrinsically strong Raman scattering molecules (called Raman reporters) to the surface of plasmon-resonant nanoparticles, binding to different epitopes on a protein biomarker was also reported to improve the detection sensitivity [17]. It is clear that the efficiency of detection was affected by the preparation steps. As the SERS effect is related to the surface roughness of the substrate [18], a new class of multi-branched nanostructures such as nanostars [19,20] and nanoflowers [21], which have a higher surface roughness than spherical particles of similar size, has been synthesized. In addition to these monometallic nanostructures mentioned above, multi-component nanostructures such as bimetallic core–shell nanoparticles [22], nanourchins [14] and satellite nanostructures [23] have also attracted increasing attention [24]. 

In this work, Ag@Au core–shell porous nanocages were synthesized through galvanic replacement reactions using an Ag nanoparticle as a self-sacrificial template. The as-prepared nanostructures exhibit excellent SERS activity, a desired property of SERS tags to develop highly sensitive SERS immunosensors (Scheme 1). In order to maximize the sensitivity of the SERS signal, the size of the Ag core and surface roughness of the porous Au shell were optimized by adjusting the amount of chloroauric acid added to the solution that contained Ag nanoparticles with diameters of 50 ± 5 nm. Using numerical differentiation (DDA), it was calculated that the enhancement of the electromagnetic field generated by the active sites around the nanopores can be more than 10 times higher than that at other sites [25], implying that the Ag@Au porous nanocage prepared in this study can provide a plentiful number of hot spots with enormously enhanced SERS activity. The excellent SERS activity of the Ag@Au core–shell porous nanocage SERS tags led to ultrahigh sensitivity, in which α-fetoprotein (AFP) could be detected down to 0.12 ng mL^-1^ without the need of further signal amplification.

## 2. Materials and Methods

### 2.1. Chemical Reagents

Silver nitrate (AgNO_3_, 99.8%) was purchased from Shanghai Chemical Reagent Co., Ltd. (China). Sodium citrate dihydrate (99.0%), ascorbic acid (99.99%), chloroauric acid tetrahydrate (99.9%), 4-mercaptobenzoic acid (4-MBA), bovine serum albumin (BSA), (3-aminopropyl) triethoxysilane (APTES, 96%) and glutaraldehyde (50% wt in H_2_O) were purchased from Aladin. Capture antibody (Ab_1_) and detection antibody (Ab_2_) of AFP (clone no. A2 and A4) were purchased from Beijing Bioss Biotechnology Co., Ltd. AFP standard solutions with concentrations from 0 to 500 ng mL^−1^ were taken from ELISA kits of AFP, which were obtained from Fujirebio Diagnostics AB (Göteborg, Sweden). Ultrapure water prepared with a Millipore water purification system (18 MΩ, Milli-Q, Millipore) was used in all assays. Phosphate buffer (PBS, 0.01 M) of various pHs was prepared by mixing the stock solutions of NaH_2_PO_4_ and Na_2_HPO_4_. The washing buffer was PBS (0.01 M, pH 7.4) containing 0.02% (w/v) Tween-20 (PBST). All other reagents were of analytical grade and used as received.

### 2.2. Synthesis of Ag Nanoparticles

Ag nanoparticles (Ag NPs) were prepared through the reduction of AgNO_3_ by ascorbic acid and sodium citrate under chloride ion induction [26]. Ag NPs of 50 ± 5 nm in diameter were obtained by using the above method. First, 1.0 mL of 34.0 mM sodium citrate and 30.0 μL of 0.1 M NaCl solution were successively added to 2.4 mL of 0.01 M AgNO_3_ solution, then mixed. After 5 minutes, the mixture was rapidly added to 46.6 mL of boiling water that contained 100.0 μL of 58.3 mM ascorbic acid, with the ascorbic acid being added 1 minute prior to the addition of the mixture. Finally, the solution was maintained at the boiling point for 1 h, then cooled to room temperature while being stirred to obtain the hydrosol of Ag nanoparticles. The obtained hydrosol was centrifuged at 10,000 rpm and the precipitates were re-dispersed in water. The Ag NPs hydrosol was filtered through a 0.22 μm filtration membrane. 

### 2.3. Synthesis of Ag@Au Core–Shell Porous Nanocages

At room temperature, 6.0 mL of the hydrosol of Ag nanoparticles was added to a glass bottle and ultrasonicated for 30 min. After that, 0.253 mM chloroauric acid solution was slowly added via a peristaltic pump while the mixture was continuously stirred for 30 min. Different volumes of chloroauric acid solution (0.5, 1.0, 1.5 and 2.0 mL) were studied. The Ag@Au core–shell porous nanocages hydrosol was filtered through a 0.22 μm filtration membrane.

### 2.4. Preparation of SERS Tags

First, 30.0 μL of 1.0 mM 4-MBA in ethanol was added to the hydrosol of the Ag@Au core–shell porous nanocages using 1.0 mL of 0.253 mM chloroauric acid aqueous solution prepared by stirring for 1 h. Then, 200.0 µL 20.0 ug mL^−1^ Ab_2_ was added to the mixture while stirring for 6 h at 4 °C. Afterward, 100.0 µL of blocking buffer containing 5% BSA and 0.02% Tween-20 was added to the mixture while stirring for 2 h, followed by aging without stirring for 12 h at 4 °C. The obtained Ag@Au core–shell porous nanocage SERS tags were centrifuged at 9000 rpm and the precipitates were dispersed in 0.01 M PBS solution. Similarly, the same procedures were used for the synthesis of Au nanoparticle SERS tags for comparison.

### 2.5. Preparation of Immuno-Sensing Chips

Quartz chips were placed into piranha solution (H_2_O_2_/H_2_SO_4_, 1:3, v/v) for 1 h, then rinsed with water and dried with nitrogen gas. The cleaned quartz chips were then placed into anhydrous ethanol containing 5.0 mM (3-aminopropyl) triethoxysilane APTES for 12 h. It was combined with APTES via O–Si–O covalent bonds to form an aminated self-assembled monolayers. The modified chips were washed thoroughly with anhydrous ethanol and dried with nitrogen gas afterwards. Polyethylene terephthalate film with an array of 3 mm diameter holes was carefully applied to the above modified quartz chip. Next, 5.0 μL of 2.5% glutaraldehyde solution was pipetted into each well and kept there for 2 h at room temperature before being rinsed with washing buffer. Afterward, 10.0 μL of Ab_1_ was pipetted into each well and incubated for 4 h at room temperature before being rinsed with water. Subsequently, the blocking buffer that was diluted 10 times was pipetted onto the chips and incubated for 1 h. The AFP immuno-sensing chips were then rinsed with washing buffer and stored in a plastic centrifuge tube in a nitrogen atmosphere.

### 2.6. Sandwich Immunoassay

The AFP solution of different concentrations was pipetted onto the immuno-sensing chips and incubated for 30 min at 37 °C. The modified chips were then rinsed with washing buffer. The Ag@Au core–shell porous nanocage SERS tags were subsequently pipetted onto the AFP-captured immuno-sensing chips and incubated for 30 min at 37 °C. Finally, the un-captured Ag@Au core–shell porous nanocages were washed away with washing buffer. The chips were then dried under nitrogen protection prior to SERS measurement.

### 2.7. Instruments and Measurements

Transmission electron microscopy (TEM) images were obtained with a JEOL JEM-2100 high-resolution transmission electron microscope, using an accelerating voltage of 200 kV. Ultraviolet-visible (UV-vis) spectra of the samples was obtained with a Shimadzu ultraviolet spectrophotometer (UV-2450). Raman spectra were recorded on a Renishaw inVia Raman spectrometer equipped with a charge coupled device (CCD) detector. The excitation lasers with a wavelength of 785 nm were used for measurements. SERS spectra were acquired using two platforms: (1) SERS spectra of 4-MBA-labeled Ag@Au core–shell nanocages, where the typical exposure time for each measurement was 10 s with one-time accumulation and the laser power was set to 10 mW; and (2) sandwich assays between the SERS tags and the immuno-sensing chips, where the typical exposure time for each measurement was 20 s with one-time accumulation and the laser power was set to 10 mW. All of the SERS spectra shown in this report were baseline-corrected, but not normalized.

## 3. Results and Discussion

### 3.1. Characterization of Ag Core–Au Shell Nanocages

Ag NPs with a diameter of 50 ± 5 nm were first synthesized with AgNO_3_, ascorbic acid and sodium citrate under chloride ion induction. The localized surface plasmon resonance (LSPR) band of the as-synthesized Ag NPs appeared at 415 nm, and the maximum absorption gradually shifted to 440 nm, accompanied by a decrease in intensity. Ag@Au core–shell porous nanocages were readily synthesized through galvanic replacement reactions using Ag NPs as the sacrificial templates. The Ag absorption band was broadened upon the increase of the amount of chloroauric acid added, owing to the formation of the Au shell on the surface of the Ag core and the superposition of their LSPR band (Figure 1). Upon continuous increase of chloroauric acid, a new band appeared in the range of 500–700 nm, presumably due to the LSPR absorption of the Au shell. The LSPR band of Ag NPs was progressively weakened, indicating that the Ag core had been gradually replaced and oxidized by HAuCl_4_. Phenomenologically, the corresponding hydrosols underwent a series of distinct color changes from yellowish green to reddish brown, purple and blue (Figure 1a–e).

As shown in Figure 2A, the nearly uniform Ag@Au core–shell porous nanocages have diameters of 50 ± 5 nm. The highly matched lattice constant of Ag (0.409 nm) and Au (0.408 nm) was too similar to be distinguished (Figure 2B) [27]. However, Au and Ag can be distinguished with a high-resolution TEM technique, owing to the different atomic weights of Au and Ag (196.97 and 106.87, respectively) [28]. The dark area corresponds to Au, whereas the bright area represents Ag. As shown in Figure 2A, the Au shell has very rough surface filled with 3–5 nm sized pores.

### 3.2. Characterization of SERS Tags

As shown in Scheme 1, the Ag@Au core–shell porous nanocages were first modified with 4-MBA through covalent bonding between Au and S. Then, the 4-MBA-decorated Ag@Au core–shell porous nanocages were modified by Ab_2_ through the reaction of Au and protein. The ultraviolet-visible (UV-vis) spectra show changes before and after the biological functionalization of Ag@Au core–shell porous nanocages (Figure 3A). The LSPR band of 50 nm Ag@Au core–shell porous nanocages was located at 450 nm (Figure 3A, curve a). After functionalization with 4-MBA and Ab_2_, the LSPR band of the nanocage SERS tags shifted to 455 nm while the LSPR band located at 270 nm appeared (Figure 3A, curve b). The above results confirmed that 4-MBA and antibodies had successfully attached to the surface of the Ag@Au core–shell porous nanocages. For comparison, Au NPs SERS tags were also synthesized (Figure 3B).

Figure 4A shows that the 4-MBA-labeled Ag@Au core–shell porous nanocage SERS tags produced main SERS peaks at 716, 839, 1076, 1181, 1486 and 1586 cm^−1^. The Raman shifts at 1586 cm^−1^ are caused by the C=C stretching vibrations of 4-MBA. The Raman shifts at 1076 cm^−1^ are caused by the C–H bending vibrations of 4-MBA. Results in Figure 4A also illustrate that the SERS signal using Ag@Au nanocage SERS tags is more than 10 times higher than that of the Au SERS tags. This suggests that Ag@Au core–shell porous nanocages can easily outperform Au nanoparticles to make an SERS immunosensor with enormously improved sensitivity. For 15 randomly selected test points on the AFP-captured immuno-sensing chips, the intensity of the Raman peak at 4-MBA at 1076 cm^−1^ was measured as shown in Figure 4B and the relative standard deviation (RSD) of the peak intensity at each point was 6.11%, indicating that the substrate had good uniformity.

### 3.3. Optimization of the Amount of Chloroauric Acid

In order to maximize the sensitivity of the SERS signal, the size of the Ag core and the pore and surface roughness of the Au shell were investigated in this study. As shown in Figure 5, the SERS intensity of adsorbed 4-MBA first enhanced and then weakened. When the amount of aqueous chloroauric acid solution (0.253 mM) added to 6 mL Ag NPs was 1.0 mL, the SERS intensity was highest. When chloroauric acid was added into the solution of Ag NPs without any other reducing agent, corrosion appeared to commence at some specific sites of the Ag core via a galvanic replacement reaction between Ag^0^ and AuCl_4_^–^. As a result, the Ag core gradually became smaller, and the Au shell with a rough surface grew. This result may be attributed to the presence of pinholes in the Au layer that act as hot spots for electromagnetic field enhancement. However, the gradual depletion of the Ag core and the disappearance of pinholes resulted in a great decrease of the Raman signal. Under optimal conditions, the SERS intensity of Ag@Au core–shell porous nanocages is more than 10 times higher than that of Ag NPs or Au NPs. The higher SERS activity of the Ag@Au porous nanocages likely comes from the synergistic interactions of two factors: (1) newly generated hot spots in the rough Au shell for stronger plasmonic coupling and (2) the retained SERS activity of the Ag core.

### 3.4. Analytical Performance

In the presence of AFP, the SERS peaks of 4-MBA from the captured SERS tags solution could be observed and the intensity increased with the AFP concentration, as demonstrated by the strongest Raman peak at 1076 cm^−1^ (Figure 6A). The calibration plot showed a good linear relationship between the SERS intensity and the logarithm of the analyte concentration within the range from 0.2 ng mL^−1^ to 500 ng mL^−1^ (Figure 6B). The calculated correlation coefficient was 0.998, as shown in Figure 6B. Even at a concentration of AFP as low as 0.2 ng mL^−1^, the SERS peaks could still be directly detected without the need for any enhancement, which was much lower than that of the Au or Ag nanomaterial-based SERS immunoassay. The limit of detection (LOD) at a signal-to-noise ratio of 3 σ (where σ is the standard deviation of signal in a blank solution) was determined to be 0.12 ng mL^−1^.

### 3.5. Reproducibility and Precision of the Immunosensor

In order to estimate the reproducibility of the immunoassay, both the intra-assay and inter-assay precision of the immunosensor were examined with 100 ng mL^−1^ AFP five times. The RSDs were 3.2% and 3.8% (Table 1), respectively, showing good precision and acceptable fabrication reproducibility. In addition, when the immunosensor was stored in dry conditions at 4 °C, over 90% of the initial response remained after a storage period of 3 weeks. These results indicated that the immunosensor had acceptable reliability and stability, and was suitable for the clinical diagnosis of protein markers.

## 4. Conclusions

A highly sensitive SERS immunoassay of biomarkers was developed in this research, which involves the generation of 4-MBA-labeled Ag@Au core–shell porous nanocages SERS tags and subsequent modification with AFP detection antibodies. The thus-obtained AFP immuno-sensing chips integrate the selectivity of immune recognition and overcome the drawbacks of low sensitivity and stability of Au or Ag nanoparticles. Furthermore, the Ag@Au core–shell nanocage SERS tags could be optimized to provide plenty of hot spots for achieving enormously enhanced SERS activity. Overall, this study illustrates that the Ag@Au core–shell porous nanocages can be used for highly sensitive SERS assays.

## 5. Patents

Lin, D.; Huang, Y.; Li, M.;Wang, S; Jin, H. Preparation of Ag@Au core–shell porous nanocages, surface-enhanced Raman spectroscopy detection probes and its applications. CHN. Patent CN 108580919 A, 28 September 2018.

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
