# Peer review of "Ag@Au Core–Shell Porous Nanocages with Outstanding SERS Activity for Highly Sensitive SERS Immunoassay"

_sensors, 2019, doi:10.3390/s19071554_

Round 1

Reviewer 1 Report

The manuscript entitled Ag@Au core-shell porous nanocages with outstanding SERS activity for highly sensitive SERS immunoassay” deals with the synthesis of Ag-Au core-shell porous nanocages and their incorporation in immune-sensing chips sensitive towards alpha-fetoprotein, giving sensors with as low detection limit as 0.12 ng/mL and a linear range of 0.2-500 ng/mL. I find the present manuscript and the data within of interest for the scientific community and I recommend publication of the manuscript in Sensors.

I have a couple of minor comments and questions that I recommend the author’s to address:

The abstract contains both the short and long names of AFP. I recommend using the long name in the abstract.

Do the authors refer to HAuCl4 in the abstract when they say chloric acid? If yes, I recommend using the correct name, e.g. chloroauric acid or tetrachloroauric acid. The same is valid for row 58 in the introduction, where the authors use the term “chlorinic acid”.

Maybe the word “superexcellent” used in the abstract for the sensor characterization is a bit too much.

What do the authors mean by “enrichment” in row 65 of the introduction?

Materials and Methods section - Could the authors elaborate the need to use 0.22 μm membranes to collect the nanoparticles since their size is only around 50 nm? How are the particles retained by the membrane?

Perhaps the synthesis method for the Au nanoparticles SERS tags used for comparison in this study could also be briefly mentioned in the Material and Methods section.

Section 2.4. and 2.5 – I recommend introducing the full names as well for APTMS, Ab2, Ab1.

Are all the sensing experiments obtained using 1 mL of 0.253 mM HAuCl4? Unless I have missed it, I recommend stating this issue in the manuscript body.

Caption of Figure 4(B) – I suggest being more specific and replacing “Intensities of SERS peaks” with “Intensities of 1076 cm-1 SERS peak”

Caption of Figure 5 refers to curve f, but I do not see any curve f on the graph.

Figures 4(A), 4(B) and 6(B) – since the y-axis is in arbitrary units, I recommend deleting the number of counts from this axis.

Figure 6(B) inset represents the logarithmic plot of SERS intensities with concentration, therefore it would help replacing “AFP” with “log AFP” in the inset x axis.

Author Response

To Reviewer 1:

Dear Sir:

Thank you very much for your valuable comments regarding our above paper. According to your opinions and suggestions, we have revised the manuscript. May I reply to your comments and show you the changes in the revision as follows:

The manuscript entitled “Ag@Au core-shell porous nanocages with outstanding SERS activity for highly sensitive SERS immunoassay” deals with the synthesis of Ag-Au core-shell porous nanocages and their incorporation in immune-sensing chips sensitive towards alpha-fetoprotein, giving sensors with as low detection limit as 0.12 ng/mL and a linear range of 0.2-500 ng/mL. I find the present manuscript and the data within of interest for the scientific community and I recommend publication of the manuscript in Sensors.

1. The abstract contains both the short and long names of AFP. I recommend using the long name in the abstract.

Response: AFP is the short name of α-fetoprotein.We have revised “AFP” to “α-fetoprotein” in page 1, line 14.

2. Do the authors refer to HAuCl4in the abstract when they say chloric acid? If yes, I recommend using the correct name, e.g. chloroauric acid or tetrachloroauric acid. The same is valid for row 58 in the introduction, where the authors use the term “chlorinic acid”.

Response: Thank you for your suggestion. We have revised “chloric acid” to “chloroauric acid” inpage 1, line 18;page 2, line 65;page 3, line 79;page 3, line 103;page 3, line 105;page 4, line 155;page 4, line 157;page 5, line 164page 5, line 176;page 7, line 215;page 7, line 216.

3. Maybe the word “superexcellent” used in the abstract for the sensor characterization is a bit too much.

Response: We have revised “superexcellent” to “excellent” inpage 1, line 18.

4. What do the authors mean by “enrichment” in row 65 of the introduction?

Response: The word “enrichment” aims at the method of obtaining a low detection limit by using ALP enzyme-induced Ag+deposition on the surface of Au NPs or multiple SERS probes binding together for signal amplification. The relevant references (Reference 16 and Reference 17) have been listed. To make it easy to understand, we have revised “enrichment” to “further signalamplification” in page 1, line 23;page 2, line 72.

5. Materials and Methods section-Could the authors elaborate the need to use 0.22 μm membranes to collect the nanoparticles since their size is only around 50 nm? How are the particles retained by the membrane?

Response: The nanoparticle collosol we prepared is uniform in diameter size about 50±5 nm with little agglomerated nanoparticle. In order to further improve the nanoparticle uniformity, the nanoparticle of 50±5 nm was retained and the agglomerated nanoparticle was removed in the solution using 0.22 μm filtering membrane. Accordingly, the sentences in page 3, line 99; page 3, line 106have also been revised.

6. Perhaps the synthesis method for the Au nanoparticles SERS tags used for comparison in this study could also be briefly mentioned in the Material and Methods section.

Response: We've written this section in page 3, line 114-115. The synthesis method of Au nanoparticle SERS tags for comparison was the same as synthesis method of Ag@Au core-shell porous nanocage SERS tags.

7. Section 2.4. and 2.5 – I recommend introducing the full names as well for APTMS, Ab2, Ab1.

Response: “(3-aminopropyl) triethoxysilane” is the full names of APTES (no APTMS), which has been revised inpage2, line 81page3, line 119.Ab1is capture antibody, and Ab2is detection antibody. We have revised this sentence in page 2, line 82.

8. Are all the sensing experiments obtained using 1 mL of 0.253 mM HAuCl4? Unless I have missed it, I recommend stating this issue in the manuscript body.

Response: Yes, you are right. We have added “using 1.0 mL of 0.253 mMchloroauric acid aqueoussolutionprepared”in page 3, line 109.

9. Caption of Figure 4(B) – I suggest being more specific and replacing “Intensities of SERS peaks” with “Intensities of 1076 cm-1SERS peak”

Response: We have added “at 1076 cm-1” in page 7, line 208.

10. Caption of Figure 5 refers to curve f, but I do not see any curve f on the graph.

Response: We used two reporter molecules (malachite green and 4-mercaptobenzoic acid) and to verify this experiment. In order that the SERS reporter molecule is consistent in the manuscript, we have replaced Figure 5 of malachite green with Figure 5 of 4-mercaptobenzoic acid. Accordingly, the sentences inpage 4, line 143;page 7, line 214;page 7, line 223;page 7, line 228have also been revised. 

11. Figures 4(A), 4(B) and 6(B) – since the y-axis is in arbitrary units, I recommend deleting the number of counts from this axis.

Response: The Y-axis is the signal strength value, which has no units, but has a linear relationship with the concentration of the analyte. Therefore, we need to retain the number of counts from this axis.

12. Figure 6(B) inset represents the logarithmic plot of SERS intensities with concentration, therefore it would help replacing “AFP” with “log AFP” in the inset x axis.

Response: In Figure 6(B)inset, we used the lg coordinate representation on the origin software instead of the lg transformation of the original data. And, we only use lg coordinate for the X-axis. Therefore, we think the x-coordinate is still AFP in the inset x axis.

Reviewer 2 Report

Comments to the author:

In this manuscript the authors reported the fabrication of a new SERS tag based on Ag@Au core-shell nanocages for the detection of α-fetoprotein through a SERS-based immunoassay. The Ag@Au nanoparticles were synthesized through a galvanic replacement methodology where Ag acts as a sacrificial template. The authors showed the SERS efficiency of the nanoparticles as a function of the thickness of Au layer. Finally, they have tested the sensing capabilities of the system in the ultradetection of α-fetoprotein through an immunoassay. Overall, this contribution presents a facile, and sensitive fabrication of a SERS tag which shows great potential in the detection of α-fetoprotein. Moreover, the same strategy can be applied for the detection of other biomolecules of interest. I would recommend the publication of this contribution, but some issues should be addressed.

The following are some questions and suggestions for improving their work:

Abstract

1.       Some acronym such as MBA and AFP are not defined.

2.       Chloric acid is not the same as chloroauric acid.

Introduction

1.       SERS needs to be defined in a proper way.

2.       SERS-based immunoassay is not defined.

3.       Comparison of SERS-based immunoassay with infrared and fluorescence spectroscopy in terms of high selectivity and resolution is not correct, it should be SERS spectroscopy.

4.       Line 4. SERS signal is enhanced 4 times in comparison with what?

5.       SERS tags are not defined.

6.       There is no introduction about the molecule that is going to be tested and its importance.

After all the introduction it is not clear the purpose of the work. What it is the interest/advantage of the combination of metallic nanoparticles with immunoassays? Why Ag@Au nanoparticles? How it is usually determined α-fetoprotein? Which are the reported limits of detection? How you quantify the SERS efficiency?

Methodology

1.       The reported sized must be express with a standard deviation not with a range of values.

2.       Ag nanoparticles of different sizes has been synthesized? Which are the amounts?

3.       Line 96. If nanoparticles are colloidally stable, why do they need 30 min. of sonication?

4.       Line 103. After the addition of 4-MBA how much time does it take until the addition of Ab2?

Results and discussion

1.       Figure 3A. Spectra b is after 4-MBA functionalization or after 4-MBA + Ab2 functionalization. Which one causes the shift?

2.       Figure 3B-The concentration of Au NPs is the same as Ag@Au nanoparticles? As the optical properties seem that the spectra of Au nanoparticles are much more diluted (0.25mM) and Ag@Au should be around 0.5mM.

3.       Figure 4. The same, does the spectra was taken using the same concentration of different nanoparticles? Which is the laser power?

4.       Figure 5. Why this study was carried out with malachite green instead with 4-MBA? Moreover, with 785nm the system is not under resonance condition, so I do not see the point in using a dye molecule.

5.       For the calculation of the limit of detection how is the signal to noise ratio determined from the SERS spectra? This limit of detection is much higher than other works in the literature that report limits of detection of 0.31 pg/mL. So what is the advantage of this methodology?

6.       When performing the limit of detection, the signal is always homogeneous regardless the analyte concentration? How many points/measurements were done per sample? Is there any mapping measurement?

7.       Line 239. A figure should be presented from this section to support the data.

Author Response

To Reviewer 2:

Dear Sir:

Thank you very much for your valuable comments regarding our above paper. According to your opinions and suggestions, we have revised the manuscript. May I reply to your comments and show you the changes in the revision as follows:

In this manuscript the authors reported the fabrication of a new SERS tag based on Ag@Au core-shell nanocages for the detection of α-fetoprotein through a SERS-based immunoassay. The Ag@Au nanoparticles were synthesized through a galvanic replacement methodology where Ag acts as a sacrificial template. The authors showed the SERS efficiency of the nanoparticles as a function of the thickness of Au layer. Finally, they have tested the sensing capabilities of the system in the ultradetection of α-fetoprotein through an immunoassay. Overall, this contribution presents a facile, and sensitive fabrication of a SERS tag which shows great potential in the detection of α-fetoprotein. Moreover, the same strategy can be applied for the detection of other biomolecules of interest. I would recommend the publication of this contribution, but some issues should be addressed. 

Abstract

1. Some acronym such as MBA and AFP are not defined.

Response: We have revised these sentences inpage 1, line 14.

2. Chloric acid is not the same as chloroauric acid.

Response: We have revised “chloric acid” to “chloroauric acid” in page 1, line 18.

Introduction

1. SERS needs to be defined in a proper way.

Response:Surface-enhanced Raman spectroscopy (SERS) is one of the modern techniques that are capable of ultrasensitive detection and quantification of molecules on or near the surface of plasmonic nanostructures. We have revised this sentence in page 1, line 43-44.

2. SERS-based immunoassay is not defined.

Response:SERS-based immunoassay is defined that SERS technique capable of ultrasensitive detection and quantification of molecules on or near the surface of plasmonic nanostructureshas great applications in immunoassay. We have added the defined of SERS-based immunoassay in page 1, line 44.

3. Comparison of SERS-based immunoassay with infrared and fluorescence spectroscopy in terms of high selectivity and resolution is not correct, it should be SERS spectroscopy.

Response: We have revised this sentence in page 2, line 45.

4. Line 4. SERS signal is enhanced 4 times in comparison with what?

Response:InReference 16, it's compared to the naked Au nanoparticles.We have revised this sentence in page 2, line 49.

5. SERS tags are not defined.

Response: According to Reference 15, SERS tags are defined by attaching intrinsically strong Ramans cattering molecules (called Raman reporters) to the surface of plasmn-resonant nanoparticles.We have added “attaching intrinsically strong Ramans cattering molecules (called Raman reporters) to the surface of plasmn-resonant nanoparticles” in page 2, line 50-52.

6. There is no introduction about the molecule that is going to be tested and its importance.

Response: α-fetoprotein (AFP) has been developed as a biomarker for the early diagnosis of liver cancer. Generally, the AFP level exceeding the threshold of 25 ng mL-1in serum is suspected to be associated with the occurrence of liver cancer or other related diseases. We have added these sentences inpage 1, line 32-35.

After all the introduction it is not clear the purpose of the work. What it is the interest/advantage of the combination of metallic nanoparticles with immunoassays? Why Ag@Au nanoparticles? How it is usually determined α-fetoprotein? Which are the reported limits of detection? How you quantify the SERS efficiency? 

Response: It is the purpose of the work that a highly sensitive SERS based-immunoassay of biomarkers has been achieved using 4-mercaptobenzoic acid and Ab2labeled Ag@Au core-shell porous nanocage tags and α-fetoprotein immuno-sensing chips.The Ag@Au porous nanocages exhibits excellent surface-enhanced Raman spectroscopy (SERS) activity due to synergetic interaction between newly generated hot spots in the rough Au shell and the retained SERS activity of Ag core easily controlled by adjusting chloroauric acid concentration. Moreover, compaired withAg NPs or Au@Ag NPs, the Ag@Au porous nanocages exhibits good biocompatibilityfor antibody labelling because of Au shell. Compaired with Au NPs, the SERS signal using Ag@Au nanocage SERS tags is more than 10 times higher than that of the Au SERS tags in Figure 4A and Figure 5.Using sandwich immunoassay,in the presence of AFP, the SERS peaks of 4-MBA from the captured SERS tags solution added could be observed and the intensity increases with the AFP concentration, as demonstrated by the strongest Raman peak at 1076 cm-1(Figure 6A). These have been mentioned in this manuscript, respectively.

Methodology 

1. The reported sized must be express with a standard deviation not with a range of values.

Response: The reported sized was 50±5 nm. We have revised these sentences in page 2, line66;page 3, line 92page 4, line 150;page 5, line 168.

2. Ag nanoparticles of different sizes has been synthesized? Which are the amounts?

Response: In fact, we've synthesized three sizes of Ag nanoparticles, such as 20 nm, 40 nm, 50 nm. According to the relationship between SERS effect and substrate size, we selected 50 nm Ag nanoparticles as the experimental material, and other materials were not relevant to this manuscript, so the sentence of “Ag nanoparticles of different sizes can be obtained by varying the amounts of AgNO3used.” has been deleted.

3. Line 96. If nanoparticles are colloidally stable, why do they need 30 min. of sonication?

Response:Collosol belongs to an unstable thermodynamic system. Sol particles can conglomerate gradually into greater size. We used ultrasonic treatment for 30 min to obtain better homogeneity.

4. Line 103. After the addition of 4-MBA how much time does it take until the addition of Ab2?

Response:It takes 1 h after the addition of 4-MBA until the addition of Ab2. We have revised this sentence in page 3, line 110

Results and discussion

1. Figure 3A. Spectra b is after 4-MBA functionalization or after 4-MBA + Ab2functionalization. Which one causes the shift?

Response: Figure 3A. Spectra b is after 4-MBA and Ab2functionalization.Accordingly, the sentence in page 6, line 184has also been revised.

2. Figure 3B-The concentration of Au NPs is the same as Ag@Au nanoparticles? As the optical properties seem that the spectra of Au nanoparticles are much more diluted (0.25mM) and Ag@Au should be around 0.5mM.

Response: The formate of absorbance (A) is calculated:

A=abc

Here, A is absorbance value; a is absorptivity (L g-1cm-1); b is optical path length (cm); c is solution concentration (g L-1).

The peak of UV-vis spectra of colloid is related to the concentration and absorptivity of the material. The peaks of UV-vis spectra between different materials cannot be used to directly compare their concentrations.

3. Figure 4. The same, does the spectra was taken using the same concentration of different nanoparticles? Which is the laser power?

Response: Using sandwich immunoassay, in the presence of AFP, the SERS peaks of 4-MBA from the captured SERS tags solution added could be observed and the intensity increases with the AFP concentration. SERS spectra of the AFP immuno-sensing chips incubated with AFP of 100 ng mL-1followed by incubating with the Ag core-Au shell nanocage SERS tags (a) and Au NP SERS tags (b), washing, and drying. The captured different SERS tagshave the same amount on the surface of immunosensor due to the same amount of AFP. The laser power is 10 mW. We have added this sentence in page 7, line 209-210.

4. Figure 5. Why this study was carried out with malachite green instead with 4-MBA? Moreover, with 785nm the system is not under resonance condition, so I do not see the point in using a dye molecule.

Response:We used two reporter molecules (malachite green and 4-mercaptobenzoic acid) and to verify this experiment. In order that the SERS reporter molecule is consistent in the manuscript, we have replaced Figure 5of malachite green with Figure 5of 4-mercaptobenzoic acid. Accordingly, the sentences in page 4, line 143;page 7, line 214;page 7, line 223;page 7, line 228have also been revised.The selection of laser takes into account the resonance effect and the fluorescence effect of biological molecules. 785 nm laser is suitable for the detection of biological materials.

5. For the calculation of the limit of detection how is the signal to noise ratio determined from the SERS spectra? This limit of detection is much higher than other works in the literature that report limits of detection of 0.31 pg/mL. So what is the advantage of this methodology?

Response:The background signal of the SERS peaks intensity was about 200 when the concentration of AFP is 0 ng mL-1. In this work, the limit of detection was obtained at a signal-to-noise ratio of 3σ (where σ is the standard deviation of background signal) according to many references, such as Biosens. Bioelectron. 2010, 25, 2657-2662. As the value of 3σ was 45, the limit of detection was calculated from a signal of 245. In Figure 6B, peak values of 245 was plug in the work curve equation, which were corresponding to concentrations of 0.12 ng mL-1. We have revised the description of 3σ in page 8, line 242The detection limit of this work was indeed higher than some literatures. However, the advantage of this work was that it used the SERS based-immunassay and did not use further signal amplification strategy, which reduced the error caused by signal transmission. The wide linear range of this work from 0.2 ng mL-1to 500 ng mL-1can include the threshold of 25 ng mL-1of AFP level in human serum.

6. When performing the limit of detection, the signal is always homogeneous regardless the analyte concentration? How many points/measurements were done per sample? Is there any mapping measurement?

Response: When performing the limit of detection, the signal is always homogeneous regardless the analyte concentration.15 points were measured for per sample, as shown in Figure 4B.Usually, there are two ways of quantitative measurement, points measurement and mapping measurement, respectively.We have used points measurements, for example Reference 14.

7. Line 239. A figure should be presented from this section to support the data.

Response: Wehave added Table 1inpage 8, line 259-260.

Round 2

Reviewer 2 Report

The authors has answered and corrected all the questions required. I would recommend the publication of this contribution,